# Factors That Influence the Purchase Intention of Andean Grains in University Students in the Peruvian Market

**DOI:** 10.3390/foods14234168

**Published:** 2025-12-04

**Authors:** Dany Yudet Millones-Liza, Elizabeth Emperatriz García-Salirrosas, Edgar Mayta-Pinto

**Affiliations:** 1Unidad de Ciencias Empresariales, Escuela de Posgrado, Universidad Peruana Unión, Lima 15102, Peru; 2Escuela Profesional de Administración, Facultad de Ciencias Empresariales, Universidad Peruana Unión, Lima 15102, Peru; 3Grupo de Investigación e Innovación para el Emprendimiento y Sostenibilidad, Universidad Nacional Tecnológica de Lima Sur, Lima 15816, Peru; egarcias@untels.edu.pe; 4Escuela Profesional de Ingeniería de Industrias Alimentarias, Universidad Peruana Unión, Carretera Arequipa Km 6, Juliaca 21100, Peru; edgar.mayta@upeu.edu.pe; 5Dirección General de Investigación, Universidad Peruana Unión (UPeU), Lima 15102, Peru

**Keywords:** consumer behavior, university students, Andean grains, purchase intention

## Abstract

Faced with the transformation of food and the rapid pace of life among university students, consumption behaviors for food products such as Andean grains have become significant topics. To better understand this issue, this study aimed to analyze the factors that influence the purchase intention of Andean grains in university students, based on the theory of planned behavior, self-identity, moral obligation, and willingness to pay more. The study recruited 900 university students, and the results report that moral obligation is the strongest predictor (β = 0.295), followed by self-identity (β = 0.293). These findings confirm the need to explore new opportunities for transforming and innovating Andean grains.

## 1. Introduction

Quinoa is a product recognized for its high-quality proteins and nutritional values. It has been recognized as an important grain that is part of the history of the Andean peoples [1,2], and is considered in some studies a superfood that is environmentally friendly and is capable of reducing the risk of various diseases [3]. One of the most notable characteristics of this Andean grain is its resilience to climate change, which is why the United Nations Food and Agriculture Organization has recognized it as an example of sustainable agriculture [4]. Its production in the Andean region is significant, with a greater emphasis on Peru, a country that boasts around 3000 varieties of crops, making it one of the world’s leading producers of quinoa [5]. Although this Andean grain has gone through various price fluctuations due to high production costs [6], due to its high nutritional value, national and international entities are promoting its use worldwide to combat malnutrition and support sustainable agriculture [7,8].

Consequently, quinoa is in high demand for export, and its production has spread to 123 countries [9,10]. According to the information above, its consumption internationally is increasing. However, what is happening in the national territory, a territory where this Andean grain is grown and where greater accessibility of consumption is assumed? To answer this question, a rapid review of the literature was carried out, where it was identified that after the arrival of the pandemic, consumers increased awareness regarding their health, which translated, in the Peruvian context, into a greater consumption of quinoa, with women and people with higher incomes adopting this purchasing behavior [11].

On the other hand, in the same Peruvian context, it was found that quinoa consumers are segmented into two groups: the first being consumers due to attitude and ethnic identity; the second group being consumers due to subjective norms and previous experience [12]. In addition, according to the research of [13], Peruvians tend to increase their consumption of quinoa every time their family income rises; in contrast, they give greater priority to investing in the purchase of basic foods such as rice, pasta, and potatoes at lower incomes. Meanwhile, other evidence from the literature establishes that, at the level of South America, Bolivia and Peru are the countries that have the highest consumption of quinoa in both rural households and urban segments [14], and despite price volatility, quinoa consumption is a constant for Peruvian producers of this food.

Until now, the nutritional and environmental advantages of Andean grains have been exposed, and it has been mentioned that the consumption of quinoa in Peru is a constant practice by the adult population; however, the consumption behavior of this superfood in the youth population is unknown and has not been identified in the literature. Specifically, studies have not considered university students, a population characterized by an accelerated pace of academic life, where studying represents a greater priority and students often limit themselves in consuming healthy foods due to having greater accessibility to fast food [15,16], or simply because healthy foods made from Andean grains are usually more expensive [17], acquiring bad eating habits during university life. Taking into account this knowledge gap, this study proposes to analyze the factors that influence the purchase intention of Andean grains among Peruvian university students. For this purpose, the factors of planned behavior, moral obligation, self-identity, and the intention to pay more are analyzed.

This study addresses the lack of knowledge regarding young university students, who are considered agents of change in dietary transitions. These students are highly likely to adopt sustainable diets that contribute to their health [18,19]. This is reinforced at the university level through the promotion of health care by universities, which coincides with the time when students shape their consumption patterns, due to their independence, social influence, and other factors that lead to decision-making in their food choices [20,21,22]. Thus, the perception of this population takes on unique relevance in the Andean grain market, since unlike other consumers, this population remains highly open to experimenting with new or rediscovered foods [23,24]. This research demonstrates how young university students can reconcile cultural identity with modern food preferences. This knowledge closes an important gap in our understanding of consumer behavior among a population that can contribute to the sustainability of the Andean grain value chain.

## 2. Literature Review and Hypothesis Development

Self-identity has been defined as the way a person perceives themselves and how they define themselves within the role of consumer, which is reflected in their choice of a product or brand, thereby projecting or approximating their own image [25,26]. According to previous research, self-identity is one of the most significant predictors of the intention to acquire products that improve or preserve health [27,28]. It is shown that when a person makes purchase decisions, they seek to make them consistent with their way of being, with their image, thus generating harmony between who they are and their purchase choices [29]. Research has shown that self-identity is an important factor in the intention to purchase healthy, organic products, as people are now more health-conscious [30,31]. This fact has been confirmed in previous research, which supports the idea that consumers with self-identity are more likely to purchase products with which they identify or describe themselves [32,33].

Specifically in university students, when nutritional knowledge is part of their identity, they are more likely to buy products that contribute to the conservation of their health, adopting a healthy eating behavior over time [34]. In addition, self-identity, being considered an outstanding part of oneself, describes characteristics with which a person defines themselves, the same ones that directly influence the planning of actions and decision-making related to consumption [35]; for example, in students, to the extent that they are oriented towards the care of their health and make this idea their own, their interest and intention to consume healthy products can be increased due to the self-identity they develop every time university facilities offer healthy options such as fruits, vegetables, or other healthy choices [27,35]. Based on what is referred to in these paragraphs, the following hypothesis is proposed:

**H1:** 
*Self-identity (SI) has a positive influence on the purchase intention (PI) of Andean grains among university students.*


Another factor that influences purchase intention is moral obligation, which refers to a person’s duties or responsibilities towards others. These duties are fundamentally based on moral principles [36]. In addition, moral obligation is a feeling that arises from social interaction. It motivates individuals to fulfill their duties of cooperation, encompassing sense of responsibility, social interaction, and philosophical principles [37,38], also referred to as the duty a consumer feels to do the right or wrong thing in a specific situation [39]. The theoretical framework of these three moral motives provides a better understanding of this fact by distinguishing the benefits of doing good for others from complying with rules and being consistent with one’s identity [40]. In this context, a consumer who maintains a sense of responsibility, translated into moral obligation, acts under principles and values that allow them to choose appropriate products that contribute to fair trade and generate personal benefits [41], since it is known that a consumer with moral obligation maintains a favorable valuation of a particular product as long as it maintains characteristics that benefit themselves and others [42]. Previous studies have shown that the theory of planned behavior has significant predictive power regarding purchase intention. However, other research shows that integrating moral obligation can increase this power even more [43].

Faced with this reality, the following hypothesis is proposed:

**H2:** 
*Moral obligation (MO) has a positive influence on the purchase intention (PI) of Andean grains among university students.*


Consumer behavior is a variable that has been measured by different models, some of which are referred to in the scientific literature, such as the behavioral perspective model [44], Hull–Spence behavior theory [45], and the theory of planned behavior [46,47]. The common denominator of the theories referred to is that they all measure consumer attitudes; this means that when a consumer maintains a positive attitude towards a particular product, this feeling significantly influences purchase intention [48]. When the consumer has a positive attitude, purchase intention increases, even more so when it comes to products that generate general well-being [49]. In general, previous and recent research has shown that consumer behavior predicts purchase intention; in this context, it has been demonstrated that the application of planned behavior theory predicts purchase intention across various cultural contexts and various products, including healthy and heritage foods [50,51,52].

Specifically, the subjective norm, being exposed to social pressure, can regulate consumer behavior. This means that when a consumer perceives that their social circle has a preference tendency, they feel pressured to adopt the same preferences, driving this fact to purchase intention [53]. A likely explanation for this fact is that a large percentage of people feel good when their environment approves of each action they take. In this context, the subjective norm produces emotions that lead to making purchase decisions [54,55]. Likewise, behavioral control is the perception of the complexity of adopting a specific behavior, which is dependent on factors such as previous experiences and anticipated obstacles [49]; therefore, every time a consumer maintains control over their purchase decisions, the probability of consuming a specific product increases, and if it is a healthy product, which generates benefits for the same consumer, this influence increases [56]. Based on these studies, the following hypotheses are proposed:

**H3:** 
*Attitude towards behavior (ATT) positively influences the purchase intention (PI) of Andean grains in university students.*


**H4:** 
*The subjective norm (SN) positively influences the purchase intention (PI) of Andean grains in university students.*


**H5:** 
*Perceived behavioral control (PBC) positively influences the purchase intention (PI) of Andean grains in university students.*


Another factor that has been analyzed to explain purchase intention is the willingness to pay more, which is defined as the monetary amount that a consumer is willing to give in exchange for a product or service, and depends on its usefulness [57]. In this regard, a background has been identified that one of the drivers of the purchase intention of products that are beneficial for health is price [47,56]. Since, for many, this represents the value of the product, this means that for consumers, a high price is a symbol of quality and positive effects [58,59]. This fact has had greater emphasis since the arrival of the pandemic, a scenario where the purchasing behavior of consumers was modified, because they stopped comparing prices and looking for discounts or reasonable offers in order to achieve their well-being; in this context, it is stated that price sensitivity influences their purchase intention [60,61]. Based on this, the following hypothesis is proposed:

**H6:** 
*The willingness to pay more (WPM) positively influences the purchase intention (PI) of Andean grains in university students.*


Based on the information presented in the previous paragraphs, Figure 1 is provided, which illustrates the proposed theoretical model.

In addition to the theoretical evidence supporting the proposed model, previous studies have been identified that provide empirical evidence of the intention to purchase healthy foods among university students (Table 1). These studies are based on different theories that analyze consumer behavior, specifically that of university students.

## 3. Materials and Methods

This research was developed through a non-experimental, cross-sectional design. For sampling, a non-probabilistic method was employed at the researchers’ convenience [68], which was selected for its accessibility and feasibility. Since this study was based on collecting data in university institutions, this sampling method allowed for obtaining a substantial sample efficiently, reaching 900 participants and thus maximizing the response rate. Similarly, non-probabilistic sampling was established because there was a defined target population, in this case, Peruvian university students [69]. The sample consisted of currently registered university students who were Peruvian adults and gave informed consent, thus promoting voluntary and anonymous participation. The study participants were 900 university students. The average age was 23.03 years, the minimum age was 18 years, and the median age was 21 years. The sample’s characteristics are stated in Table 2.

### Scales

For data collection, a survey was used, which was disseminated through social networks, shared with teachers for distribution of the questionnaire, and administered via the Google Forms platform. To measure the variables, the factors of attitude, subjective norm, perceived behavioral control, moral obligation, and purchase intention were taken into account, all of which consisted of 4 items validated and applied by Duran et al. [70]. Regarding the intention to pay more metric, it consisted of two items, taken and adapted from [71]. Self-identity was measured using 4 items [72]. Each of the items was formulated to be measured on a 1–5 Likert scale, where 1 represented ‘totally disagree’ and 5 meant ‘totally agree’. To analyze the intention to pay more and self-identity, the instruments were translated through the process of back-translation by a native speaker who was not affiliated with this research. Afterward, the scales were subjected to semantic validity testing with a representative sample of 06 participants.

## 4. Results

### 4.1. Convergent Validity

The results of the analysis demonstrate that the model exhibits satisfactory convergent validity across all its constructs. The factor loadings of all indicators exceed the recommended minimum threshold of 0.70 [73], ranging from 0.836 for SN2 to 0.935 for WPM2. According to [74], loads greater than 0.70 indicate that a construct explains more than 50% of the variance of an indicator, which is considered satisfactory for establishing convergent validity. The constructs that present the highest loads are willingness to pay more (WPM), with loadings ranging from 0.925 to 0.935, purchase intention (PI), with loadings ranging from 0.890 to 0.919, and moral obligation (MO), with loadings ranging from 0.855 to 0.903.

In terms of composite reliability, all constructs exceed the minimum recommended value of 0.70 [75], with ranges between 0.727 for subjective norms (SN) and 0.863 for willingness to pay more (WPM). These results indicate high internal consistency of the indicators within each latent construct. Additionally, all constructs have Average Extracted Variance (AVE) values above the critical threshold of 0.50 [74], with ranges between 0.561 for subjective norms and 0.823 for self-identity. This confirms that each construct explains more than 50% of the variance of its respective indicators, fulfilling the convergent validity criterion established by Hair et al. [76].

The three criteria evaluated for convergent validity are consistently met in the proposed model, indicating that the indicators of each construct converge adequately to measure the underlying theoretical concept [77]. This empirical evidence supports the strength of the measurement instrument used and provides confidence in the model’s ability to capture the factors that influence the purchase intention of Andean grains in Peruvian university students.

Since all data were collected through self-administered questionnaires from the same respondents at a single point in time, there was a potential risk of common method bias (CMB) [78] To assess the presence of CMB, two complementary tests were conducted following established guidelines in PLS-SEM research [73].

First, Harman’s single-factor test was performed by loading all measurement items (n = 27) into an exploratory factor analysis (EFA) without rotation. The results revealed that the first unrotated factor explained 60.24% of the total variance, which exceeds the traditional 50% threshold suggested by Podsakoff et al. [79]. However, recent methodological research indicates that this criterion may be overly conservative, particularly in studies examining theoretically related constructs where conceptual overlap is expected [80,81].

To provide a more robust assessment of CMB, a full collinearity test was conducted by examining the variance inflation factors (VIFs) of all predictor constructs in the inner structural model, as recommended by Kock [82]. The VIF values for the constructs predicting purchase intention ranged from 2.457 (subjective norms) to 4.696 (self-identity), with a mean VIF of 3.670. While five of the six constructs exceeded the conservative threshold of 3.3, all VIF values remained well below the critical threshold of 10, which is considered indicative of severe collinearity that would signal problematic common method bias [82,83]. VIF values between 3.3 and 5.0 typically reflect moderate collinearity that can be attributed to theoretically meaningful relationships among constructs rather than methodological artifacts [84].

The moderate VIF values observed are consistent with the theoretical framework, as constructs such as attitude, moral obligation, self-identity, and perceived behavioral control are conceptually related within the theory of planned behavior and its extensions. These constructs share common theoretical foundations but measure distinct psychological mechanisms [85]. Additionally, the satisfactory discriminant validity demonstrated through the Fornell–Larcker criterion (Table 2) provides empirical evidence that the constructs are sufficiently distinct, further supporting that common method variance does not pose a substantial threat to the validity of our findings [81].

Moreover, several procedural remedies were implemented during data collection to minimize CMB, including the use of validated scales established in the literature, clear and concise item wording, assurances of anonymity and confidentiality, and counterbalancing of question order [79]. Taken together, these results suggest that while some method variance may be present, it does not significantly compromise the substantive findings of this study. The combination of acceptable VIF values, strong discriminant validity, and methodological precautions provides confidence in the robustness of the structural relationships identified in the model. (See Table 3).

### 4.2. Discriminant Validity

The results of the discriminant validity analysis demonstrate that the Fornell–Larcker criterion is satisfactorily met for all the constructs in the model. The values on the diagonal, which represent the square root of the AVE, range between 0.853 for the subjective norm (SN) and 0.929 for willingness to pay more (WPM). These values are consistently higher than all correlations between constructs in their respective rows and columns, confirming that each construct shares more variance with its own indicators than with other constructs in the model [77].

The highest correlations between constructs are observed between perceived behavioral control (PCB) and attitude (ATT), with a value of 0.868, followed by the correlation between moral obligation (MO) and attitude (ATT), with 0.886. Despite these relatively high correlations, which are theoretically expected given that these constructs are conceptually related, the values of the square root of the AVE (0.868 for PCB, 0.906 for ATT, and 0.886 for MO) are equal to or exceed these correlations, maintaining the discriminant validity of the model [75] (see Table 4).

Empirical evidence confirms that all the constructs of the model have adequate discriminant validity, indicating that each latent variable captures unique phenomena not represented by other constructs in the model [86]. This conceptual and empirical distinction between the constructs strengthens the validity of the theoretical model proposed to explain the purchase intention of Andean grains in Peruvian university students.

#### 4.2.1. Hypothesis Testing

The structural equation model estimated by PLS-SEM demonstrates substantial predictive power, explaining 82.6% of the variance in the purchase intention of Andean grains (R^2^ = 0.826), which is considered a significant effect according to Cohen’s (1988) [87] criteria. The results reveal that moral obligation (MO) emerges as the strongest predictor with a path coefficient of 0.293 (*p* < 0.001), suggesting that the sense of responsibility towards the consumption of traditional Peruvian products constitutes an important factor in the purchase decision. It is followed by perceived behavioral control (PCB) with β = 0.242 (*p* < 0.001) and willingness to pay more (WPM) with β = 0.191 (*p* < 0.001), indicating that the perception of ease of access and willingness to pay a premium price significantly influence purchase intention.

On the other hand, attitude (ATT) presents a moderate positive relationship (β = 0.097, *p* = 0.002). Interestingly, subjective norms (SNs) show a significant negative relationship (β = −0.106, *p* < 0.001), suggesting that external social pressure may act as an inhibitor of purchasing behavior in this specific context, possibly due to competing social influences or perceptions about modernity versus tradition in youth consumption patterns among Peruvian university students [88]. All indicators have factor loadings greater than 0.70, ranging between 0.848 and 0.935, confirming the convergent validity of the model and providing solid empirical evidence for the structural relationships identified (see Figure 2).

Likewise, the results shown in Table 3 of the hypothesis test conducted via bootstrapping (5000 resamples with bias-corrected confidence intervals) reveal that all the structural relationships proposed in the model are statistically significant, which allows accepting the six hypotheses raised. Student’s t-values far exceed the critical threshold of 1.96 for a 95% confidence level, ranging from 3.122 to 6.963, providing robust evidence of empirical support for each causal relationship. Importantly, all 95% confidence intervals exclude zero, reinforcing the statistical significance and practical relevance of the findings.

H2 (MO → PI) presents the most significant statistical support with a t-value of 6.963, confirming that moral obligation is a strong predictor of purchase intention with a coefficient of 0.295 (95% CI [0.215, 0.379], *p* < 0.001). The confidence interval indicates that we can be 95% confident that the true population parameter for this relationship lies between 0.215 and 0.379, demonstrating a consistent positive effect. H6 (WPM → PI) shows a t-value of 6.486 with a positive coefficient of 0.242 (95% CI [0.172, 0.319], *p* < 0.001), indicating that willingness to pay more has a significant and stable influence on purchase intention. H1 (SI → PI) yields a t-value of 6.049 with a positive relationship of β = 0.293 (95% CI [0.183, 0.383], *p* < 0.001), indicating that greater self-identity as a conscious consumer significantly increases the purchase intention of Andean grains, with a confidence interval suggesting a robust effect.

The hypotheses with moderate statistical strength, though equally significant, are H5 (PCB → PI), with t = 5.165 and a positive coefficient of 0.181 (95% CI [0.114, 0.252], *p* < 0.001), suggesting that perceived behavioral control positively influences purchasing behavior with a consistent effect size, and H4 (SN → PI), with t = 4.596 and a negative relationship of β = −0.106 (95% CI [−0.155, −0.061], *p* < 0.001), indicating that subjective norms act as inhibitors of purchase intention. The negative confidence interval for subjective norms confirms that this inhibitory effect is stable across the population. H3 (ATT → PI) presents a t-value of 3.122 with β = 0.097 (95% CI [0.035, 0.157], *p* = 0.002), confirming that attitude is positively but moderately related to purchase intention. Although this effect is the smallest among the significant predictors, the confidence interval excludes zero, supporting its practical relevance.

The narrow confidence intervals observed across all path coefficients (average width of 0.12) reflect the precision of the estimates and the stability of the relationships, which is attributable to the large sample size (n = 900) and the robustness of the bootstrapping procedure [57]. The standard deviation of all estimates is low (0.024–0.048), further indicating stability in the coefficients estimated through the resampling procedure (see Table 5). These findings provide robust empirical support for the proposed theoretical model, with all hypothesized relationships demonstrating both statistical significance and substantive meaningfulness, as evidenced by the confidence intervals and effect sizes [73].

#### 4.2.2. Multi-Group Analysis by Gender

To address potential gender differences in the structural relationships of the model, a multi-group analysis (MGA) was conducted using PLS-SEM with a permutation test (5000 permutations), comparing the path coefficients between male and female university students. Table 6 presents the results of this analysis, examining whether gender moderates the relationships between the constructs and purchase intention of Andean grains.

The permutation test results indicate that there are no statistically significant differences between genders in any of the structural relationships examined (all *p*-values > 0.05). This finding suggests that the mechanisms influencing purchase intention of Andean grains operate similarly for both male and female university students, despite some observable differences in the magnitude of the path coefficients.

Although not statistically significant, some noteworthy patterns emerge from the descriptive comparison of path coefficients. For H1 (SI → PI), self-identity shows a numerically stronger positive influence among females (β = 0.363) compared to males (β = 0.227), with a difference of −0.136 (*p* = 0.172). Similarly, for H3 (ATT → PI), females exhibit a higher coefficient (β = 0.141) than males (β = 0.046), with a difference of −0.095 (*p* = 0.108), suggesting that attitude may play a somewhat more important role in women’s purchase intentions, though this difference does not reach statistical significance.

Conversely, H2 (MO → PI) reveals that males present a numerically higher coefficient (β = 0.357) compared to females (β = 0.244), with a difference of 0.113 (*p* = 0.165), indicating that moral obligation may have a slightly stronger effect on male consumers. Similarly, H6 (WPM → PI) shows males with a higher willingness to pay more coefficient (β = 0.282) than females (β = 0.196), with a difference of 0.086 (*p* = 0.259).

For H5 (PCB → PI) and H4 (SN → PI), the differences between genders are minimal. Perceived behavioral control shows nearly equivalent effects for both groups (males: β = 0.162; females: β = 0.191; difference = −0.029, *p* = 0.705), while subjective norms present negative coefficients for both genders, with a small difference (males: β = −0.082; females: β = −0.126; difference = 0.044, *p* = 0.390).

These findings suggest that the theoretical model proposed to explain purchase intention of Andean grains is robust across gender, with the same psychological and behavioral mechanisms operating similarly for both male and female Peruvian university students. From a practical standpoint, this implies that marketing strategies for Andean grains do not necessarily need to be gender-differentiated, as the key drivers of purchase intention (moral obligation, self-identity, willingness to pay more, perceived behavioral control, attitude, and subjective norms) function equivalently across gender groups. This gender invariance strengthens the generalizability of the model and suggests that interventions to promote Andean grain consumption can be designed with a unified approach for university students, regardless of gender.

## 5. Discussion

This study aimed to analyze the factors that influence the purchase intention of Andean grains in university students. The influence of self-identity on the purchase intention of these products has been demonstrated; this means that young people who identify as conscious consumers of healthy and/or traditional Peruvian products will have a greater intention to purchase Andean grains. In support of these findings, previous research reports that when university students are aware of their poor eating habits and decide to change their food decisions, they have greater intention to purchase products that preserve their health [26], and in the case of consumers who manage to identify firmly with health awareness, they are more likely to make purchases of products that align with this identity [33]. Self-identity represents a consumption pattern with which the consumer describes themself, and is a factor that could prevent inappropriate consumption behavior [89]. Our results can be compared with studies in other contexts, where it has been found that in countries such as South Africa, China, Malaysia, and Indonesia, self-identity predicts the intention to purchase organic products, as consumers with a strong sense of self-identity have a high tendency to align their purchasing decisions with their personal values and self-perception [29,33,90]. This means that self-identity has become a universal factor acquired by today’s consumers, and although it does not have the same intensity in all contexts, its effect is a cross-cultural constant.

Likewise, it has been shown that moral obligation positively influences purchase intention in the study population. This finding translates into a sense of responsibility towards the consumption of traditional Peruvian products and the support of local producers that university students have, causing the purchase intention of Andean grains to increase. A study that coincides with these results states that moral obligation stimulates the purchase intention of a specific product when it presents benefits when consumed, and the explanatory power of moral obligation adequately describes purchase intention [91,92]. One possible explanation is that consumers are putting pressure on companies to be responsible to society, which allows them to feel morally obliged to buy healthy products from local entrepreneurs, as they think it is the right thing to do according to their conscience [93,94,95]. This finding is reinforced by other studies establishing that, in countries such as Mexico, the Netherlands, China, India, and Taiwan, moral obligation is an important factor in consumers’ intention to purchase healthy products because they feel committed to sustainability, ecology, and health [96,97,98,99].

On the other hand, when evaluating the theory of planned behavior, it has been identified that attitude is a factor that positively influences the purchase intention of Andean grains; therefore, a favorable evaluation towards the consumption of Andean grains (nutritional benefits, flavor, cultural value) will allow an increase in the purchase intention of these products. Studies that support the results of this research establish that a positive attitude towards a product or service significantly influences purchase intention [95,100,101]. Regarding the subjective norm, the results indicate that, in the study population, the perception of people who are part of consumers’ social circles (family, friends, referents) alters purchase intention; a study that coincides with these findings establishes that much of consumers’ behavior is the result of social pressure, thus configuring the behavior to be the same [102,103,104]. To be more specific, in the university population, studies indicate that young people value their autonomy very much, above external or family influences; therefore, their rejection of social pressure is a response to their independence. In this population, there is a high possibility that there are economic limitations that would make the subjective norm a burden, thus generating the negative relationship with the purchase intention of Andean grains [105,106].

Within the same framework, this research provides enough evidence that demonstrates that perceived behavioral control positively influences the purchase intention of Andean grains; in other words, young people who perceive greater ease of access, economic availability, and knowledge about where to acquire Andean grains will show greater purchase intention, and the development of products that contribute to well-being remains a constant concern at any age and allows for new behavior [107,108]. Another study that supports this conclusion establishes that the components that are part of the theory of planned behavior, including perceived behavioral control, show high predictability for purchase intention [104,109], and so it is affirmed that perceived behavioral control influences consumer purchase intention, regardless of whether it is natural, healthy, ecological, or due to other products [110,111]. However, when self-identity was analyzed, the results reported that it decreases the purchase intention of Andean grains. To understand this behavior, [112,113] explain that sometimes young people resist traditional norms and seek modernity, innovation, and/or differentiation, or even prefer products that are positioned in the market or represented by famous people (influencers). The reality embodied regarding the behavior of university students is a response to globalization, an environment where traditional eating patterns are no longer attractive and they are in search of processed products, or products that are mostly accepted in their environment whenever they carry out social activities [113,114].

Likewise, this study demonstrates that the willingness to pay more influences the purchase intention of university students; this result suggests that even when the price exceeds the traditional cost of the market, university students are willing to pay more to acquire healthy food options [115]. A possible explanation for this could be that the attributes of the products and the motivation to consume them affect the willingness to pay [116,117]. Likewise, the results of this research coincide with previous studies that establish that the willingness to pay a premium price affects purchase intention and assumes a crucial role in the formation of purchase intention [118,119]. Although some cognitive factors may influence the price, it is necessary to take into account that cognitive capacity does not eliminate purchase intention. However, it could alter it [120].

## 6. Conclusions

Subsequently, it has been found that moral obligation predicts purchase intention. From this, it can be understood that the study population feels a moral duty to support local products and maintains the intention to preserve the consumption of Andean grains (β = 0.295, *p* < 0.001). Following this, it has been demonstrated that self-identity is the factor that positively predicts the intention to purchase Andean grains with the lowest percentage (β = 0.293, *p* < 0.001), which means that students identify with the consumption of Andean grains. This result suggests that educational institutions should join forces with the government to develop campaigns that connect consumers with their own cultural identity, also awakening in them the decision to support rural communities. Additionally, it is recommended to promote marketing campaigns focused on appealing to social responsibility and cultural identity, a dual approach that focuses on extrinsic ethical motivations and intrinsic ethical motivations.

Additionally, attitude and behavior have been found to be weak predictors of the intention to purchase Andean grains (β = 0.097, *p* < 0.001; β = 0.181, *p* < 0.002). This means that the availability of resources and opportunities facilitate students’ ability to translate their attitudes and intentions into concrete decisions. In this context, it is recommended that universities prioritize the availability of Andean grains at sales outlets near and within the institution, promote innovation with these grains, and encourage new ventures associated with their consumption.

Similarly, this study has shown that subjective norms have a negative effect on purchase intention (β = −0.106, *p* < 0.001). From this result, it can be deduced that young university students may resist external recommendations whenever they feel that their autonomy is at risk. Therefore, it is recommended to mitigate campaigns that reinforce young people to adopt predetermined behaviors. To this end, advertising messages that say “you should consume” should be replaced with other terms that empower autonomous decision-making, such as “discover what it means for you.”

Another predictor of the intention to purchase Andean grains is the intention to pay more (β = 0.242, *p* < 0.001). This means that a percentage of participants value these products and are willing to pay a higher price. This result also confirms that there is a differentiated market that values quality, trade, and the production of these grains, suggesting that companies should establish market segmentation strategies in order to serve price-sensitive consumers and value-oriented consumers.

### 6.1. Implications

This study shows that self-identity significantly predicts purchase intention. This provides an opportunity for educational institutions to reinforce students’ sense of identity and culture strategically. This finding aligns with previous research demonstrating that education supports students in maintaining a connection with their identity and encourages them to expand their knowledge of their heritage [121,122]. In this context, universities could implement strategies that include academic activities, such as fairs and projects, which incorporate these products to promote healthy and conscious eating habits. These activities instill a fair and accurate appreciation of the products [123].

On the other hand, identifying that moral obligation predicts purchase intention leads to the assertion that, in the study population, there is a latent ethical sensitivity. It is therefore important for the university to activate and reinforce this moral dimension in order to connect purchasing actions with social consequences such as support for the regional economy and contribution to a sustainable agricultural system, as it is precisely in this scenario that educational institutions could be a channel for acquiring experiences that can materialize moral obligation into practical action. This implication responds to evidence from previous studies that establish that educational institutions have the capacity to implement educational programs that involve action-oriented experiential learning, thus promoting rational and adequate food consumption and awareness of social, environmental, and economic consequences [124,125]. This means that decision-makers could encourage students to develop a sense of social and cultural responsibility while promoting innovative skills in these foods and supporting the preservation of this culinary heritage in the face of globalization.

Similarly, this research provides evidence of the applicability of the theory of planned behavior to traditional foods, such as Andean grains. This finding has important practical implications for universities that promote educational programs aimed at changing students’ beliefs about Andean grains and the nutritional and academic benefits of these versatile products in desserts, beverages, and more. Universities could also provide students with a social experience in which opinion leaders promote the consumption of Andean grains, making these grains available and accessible on campus to reinforce this message. Previous studies support these implications, arguing that universities have the capacity to influence consumer behavior toward healthy foods [126,127].

### 6.2. Limitations and Future Research

This study has demonstrated the behavior of the university consumer, specifically adopting non-probability sampling at the researchers’ convenience, which was used for its accessibility, feasibility, and because there was a defined target population [69]. Selection bias is acknowledged given that respondents may have answered based on their interest in or prior knowledge of Andean grains. This means positive attitudes and purchase intentions may have been overestimated. Additionally, this sampling method has limitations, such as the inability to directly extrapolate to the general population.

Additionally, since it is a cross-sectional study, perceptions were evaluated at a single point in time. This limits the ability to identify reverse causality over time. Furthermore, capturing perceptions and behaviors at a single point in time did not allow for observation of the evolution of relationships between analyzed variables. Future studies could therefore focus on longitudinal designs, such as panel studies, which re-evaluate the perceptions and behaviors of the same participants at different points in time. Similarly, longitudinal studies with experimental components are recommended to combine temporal follow-up with experimental manipulations, which can increase internal and external validity [128].

Another limitation of this study is that since it contains quantitative research that collects data through a survey, there is a risk of bias if university students have responded to the items according to what is socially desirable; to mitigate this bias, future studies could conduct analyses through qualitative research focused on exploring the motivations and barriers to the purchase of Andean grains.

Finally, this research has shown that, among university students, planned behavior theory combined with moral obligation, intention to pay more, and self-identity are predictors of the intention to purchase Andean grains. The study found no difference in perceptions between male and female groups. It is important to note that, although the study participants are from three regions of Peru, the findings cannot be generalized because Peru is known for maintaining a strong cultural heritage, including through Andean grains, which are considered part of the country’s traditional culinary heritage. Therefore, future studies could be applied to contexts without an Andean heritage to analyze replicability.

## Figures and Tables

**Figure 1 foods-14-04168-f001:**
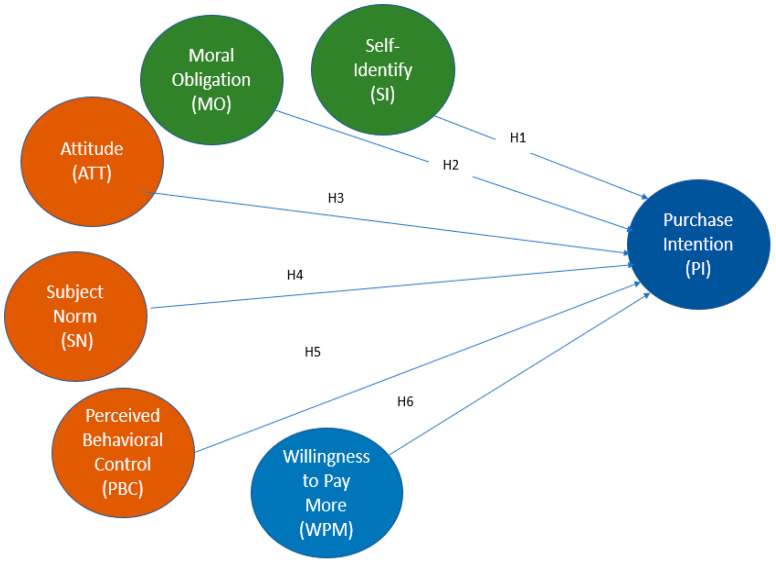
Theoretical model.

**Figure 2 foods-14-04168-f002:**
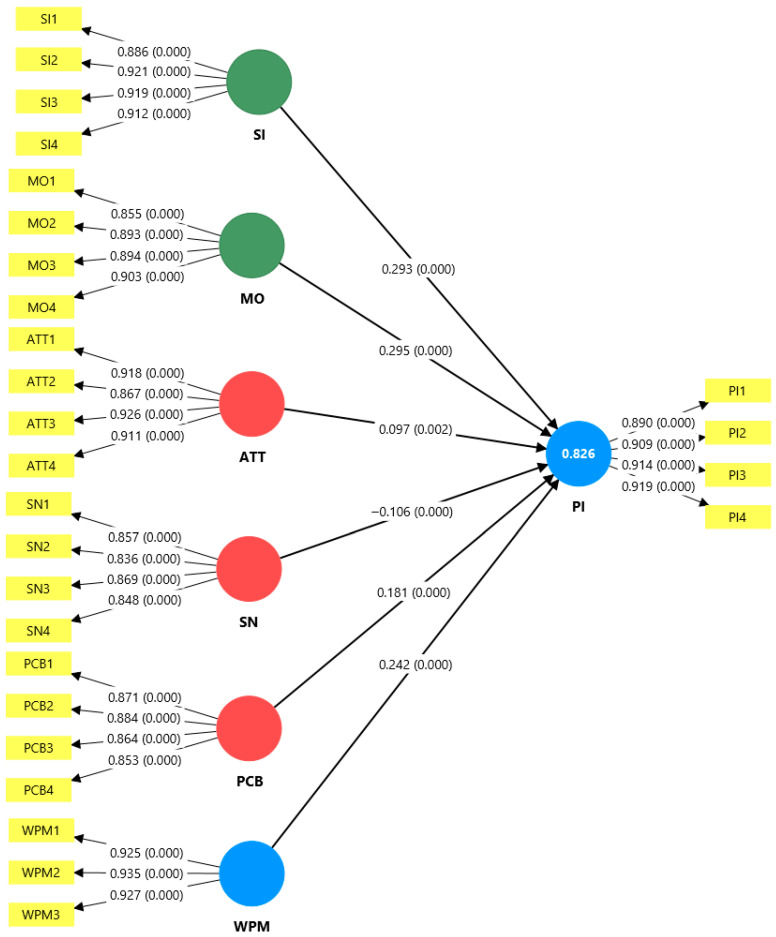
Structural model.

**Table 1 foods-14-04168-t001:** Synthesis of previous studies.

No.	Country	Theory	Results	Sample Size	Authors
1	Indonesia	Theory of Planned Behavior (TPB)	H1: Attitude → Purchase Intention: β = 0.466, t = 5.429, *p* = 0.000 H2: Subjective Norms → Intention: β = 0.222, t = 2.725, *p* = 0.007	123	[62]
2	China	Extended TPB	Health Awareness → Intention: β = 0.210, t = 2.152, *p* = 0.032 Pricing Policy → Intention: β = −0.306, t = 2.519, *p* = 0.012 Consumer Confidence → Intention: β = 0.142, t = 1.142, *p* = 0.254	335	[63]
3	Portugal	Extended TPB	Green Buying Attitudes → Intention: β = 0.269, t = 4.283, *p* < 0.001 Subjective Norms → Intention: β = −0.058, t = −1.672, *p* = 0.095 Perceived Behavioral Control → Intention: β = 0.099, t = 2.835, *p* = 0.005	432	[64]
**4**	Malaysia	Theory of Planned Behavior (TPB)	Environmental Attitude → Purchase Intention: β = 0.549, *p* < 0.001 Subjective Norms → Purchase Intention: β = 0.044, *p* > 0.05 Perceived Behavioral Control → Purchase Intention: β = 0.199, *p* > 0.05	361	[65]
**5**	Taiwan	Value–Attitude–Behavior (VAB)	Interest in Healthy Foods → Purchase Intention: β = 0.300, *p* < 0.001 Orientation Health → Purchase Intention: β = 0.693, *p* < 0.001 Signal for Action → Purchase Intention: β = −0.032, *p* > 0.05 Self-efficacy → Purchase Intention: β = 0.065, *p* > 0.05	213	[66]
**6**	Peru	Theory of Planned Behavior (TPB)	Sensory Appeal → Intention: β = 0.339, *p* < 0.001 Health Awareness → Intention: β = −0.296, *p* < 0.001 FOP Label Perception → Intention: β = −0.237, *p* < 0.001	361	[67]

**Table 2 foods-14-04168-t002:** Demographic characteristics (*n* = 900).

Sex	
Female	485
Male	415
Place of origin	
Coast	494
Mountain	347
Jungle	59
Person with whom you live	
Parents	450
Relatives	218
A friend	24
Alone	208

**Table 3 foods-14-04168-t003:** Convergent validity of the model.

Variable	Code	Loadings	Cronbach’s Alpha	Composite Reliability (rho_a)	Average Variance Extracted (AVE)	VIF
ATT	ATT1	0.918	0.927	0.948	0.821	3.427
	ATT2	0.867				
	ATT3	0.926				
	ATT4	0.911				
MO	MO1	0.855	0.909	0.936	0.786	3.762
	MO2	0.893				
	MO3	0.894				
	MO4	0.903				
PCB	PCB1	0.871	0.891	0.924	0.754	4.038
	PCB2	0.884				
	PCB3	0.864				
	PCB4	0.853				
PI	PI1	0.89	0.929	0.949	0.824	*
	PI2	0.909				
	PI3	0.914				
	PI4	0.919				
SI	SI1	0.886	0.93	0.95	0.827	4.696
	SI2	0.921				
	SI3	0.919				
	SI4	0.912				
SN	SN1	0.857	0.875	0.914	0.727	2.457
	SN2	0.836				
	SN3	0.869				
	SN4	0.848				
WPM	WPM1	0.925	0.921	0.95	0.863	3.638
	WPM2	0.935				
	WPM3	0.927				

Note: VIF = Collinearity statistic (VIF). * PI does not predict any other variable in the model, therefore it does not have a VIF in the internal structural model.

**Table 4 foods-14-04168-t004:** Discriminant validity with the Fornell–Larcker criterion.

	ATT	MO	PCB	PI	SI	SN	WPM
ATT	0.906						
MO	0.639	0.886					
PCB	0.828	0.638	0.868				
PI	0.696	0.834	0.724	0.908			
SI	0.613	0.832	0.656	0.846	0.910		
SN	0.655	0.645	0.719	0.626	0.664	0.853	
WPM	0.621	0.772	0.660	0.825	0.828	0.635	0.929

**Table 5 foods-14-04168-t005:** Hypothesis testing.

H	Path	Original Sample (O)	Sample Mean (M)	Standard Deviation (STDEV)	T-Statistics (|O/STDEV|)	*p*-Values	95% CI	Decision
H1	SI -> PI	0.293	0.294	0.048	6.049	0.000	[0.183, 0.383]	Accepted
H2	MO -> PI	0.295	0.295	0.042	6.963	0.000	[0.215, 0.379]	Accepted
H3	ATT -> PI	0.097	0.097	0.031	3.122	0.002	[0.035, 0.157]	Accepted
H4	SN -> PI	−0.106	−0.106	0.024	4.396	0.000	[−0.155, −0.061]	Accepted
H5	PCB -> PI	0.181	0.181	0.035	5.165	0.000	[0.114, 0.252]	Accepted
H6	WPM -> PI	0.242	0.241	0.037	6.486	0.000	[0.172, 0.319]	Accepted

Note: Confidence intervals were calculated using bias-corrected and accelerated (BCa) bootstrapping with 5000 resamples. All confidence intervals exclude zero, confirming the statistical significance of the relationships. Hypotheses were tested using two-tailed tests at an α = 0.05 significance level.

**Table 6 foods-14-04168-t006:** Multi-group analysis by gender using permutation test.

H	Path	Mean (Male)	Mean (Female)	Path Difference (Male–Female)	Permutation *p*-Value	Decision
H3	ATT → PI	0.046	0.141	−0.095	0.108	No significant difference
H2	MO → PI	0.357	0.244	0.113	0.165	No significant difference
H5	PCB → PI	0.162	0.191	−0.029	0.705	No significant difference
H1	SI → PI	0.227	0.363	−0.136	0.172	No significant difference
H4	SN → PI	−0.082	−0.126	0.044	0.390	No significant difference
H6	WPM → PI	0.282	0.196	0.086	0.259	No significant difference

Note: Path differences calculated as male coefficient minus female coefficient. Significant differences at *p* < 0.05 level.

## Data Availability

The original contributions presented in the study are included in the article; further inquiries can be directed to the corresponding author.

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
