# Peer review of "Factors That Influence the Purchase Intention of Andean Grains in University Students in the Peruvian Market"

_foods, 2025, doi:10.3390/foods14234168_

Round 1
Reviewer 1 Report
Comments and Suggestions for Authors
The manuscript explores the factors influencing university students’ purchase intentions toward Andean grains in Peru, using an extended Theory of Planned Behavior (TPB) model that includes self-identity, moral obligation, and willingness to pay more. The research question is relevant in the context of sustainable and healthy food consumption trends, particularly among youth. The study employs a survey-based approach (n=450) and Partial Least Squares Structural Equation Modeling (PLS-SEM) for data analysis.
However, while the topic is of interest, the manuscript in its current form requires major revision. Key theoretical assumptions are not adequately explained, particularly the negative relationships found between self-identity, subjective norms, and purchase intention. Methodologically, the use of non-probability sampling and cross-sectional design limits the generalizability and causal inference. Furthermore, several constructs lack cultural adaptation or cross-validation, and the discussion section does not sufficiently address the counterintuitive findings.
Comments to the Author:
Question 1:The finding that self-identity and subjective norms negatively influence purchase intention contradicts most prior literature. The authors should provide stronger theoretical grounding (e.g., identity conflict, modernity vs. tradition) to explain this anomaly. A simple mention of “youth seeking modernity” is insufficient.
Question 2:The use of convenience sampling undermines the representativeness of the findings. Please clarify why this method was chosen, and include a discussion on selection bias and limitations to external validity.
Question 3:The cross-sectional design limits the ability to infer causality. The authors should explicitly acknowledge this limitation and suggest longitudinal or experimental designs for future research.
Question 4:The introduction does not adequately justify the focus on university students or their unique behavioral characteristics.
Question 5:The literature review lacks a systematic synthesis of existing studies on healthy food purchase intentions, particularly among youth.
Question 6:All data were collected via self-reported questionnaires. A Harman’s single-factor test or latent method factor approach should be conducted to assess common method variance.
Question 7:The theoretical model figure lacks clear labeling of hypothesized paths, reducing its explanatory value.
Question 8:Data collection via social media and teacher distribution may lead to common method bias and sample homogeneity.
Question 9:Potential confounding variables (e.g., income, field of study) were not controlled for, affecting causal inference.
Question 10:The educational implications proposed in the conclusion are not directly derived from the results and lack empirical support.
Question 11:The cross-cultural applicability of the findings, especially in other Latin American or developing contexts, is not discussed.
Question 12:Moral obligation is the strongest predictor, but its operationalization lacks theoretical depth and clarity.
Question 13:The discussion does not sufficiently compare findings with similar studies from Asia or Africa, limiting its global relevance.
Question 14:The study did not analyze differences by gender or region, missing insights into segment-specific behaviors.
Question 15:The discussion section does not sufficiently interpret the negative effects of self-identity and subjective norms. A deeper psychological or sociocultural analysis is needed.
Question 16:Add a limitations subsection in the Discussion to clearly outline the study’s shortcomings.
Question 17:The policy and intervention implications are underdeveloped. Please elaborate on how these findings could inform university health programs, food marketing strategies, or public policy in Peru.
Question 18:The conclusions are too vague and lack specific, actionable recommendations for policy or marketing.
Question 19:Reference formatting is inconsistent; some entries lack DOIs or page numbers(e.g., [15]).
Question 20:Please proofread the manuscript for language and grammar issues, especially in the Introduction and Discussion sections.
Question 21:Consider renaming “willingness to pay more” to “premium price willingness” for clarity and academic consistency.
Question 22:there are two Table 1 in the manuscript. And Figures and tables should be reformatted for clarity and consistency with journal standards.
Author Response
Ref.: Manuscript ID: foods-3895916
Title: Factors that influence the purchase intention of Andean grains in university students in the Peruvian market
Dear Reviewer 1,
Thank you very much for your valuable comments, which have been extremely helpful in improving our manuscript. We sincerely appreciate the time and effort you have dedicated to reviewing our work.
We have carefully addressed all your suggestions and made the necessary revisions accordingly. We hope that this revised version meets the expected standards and is now suitable for publication in Foods.
A detailed response to each of your comments is provided below.
Thank you again for your time and consideration.
Best regards,
Authors
|
Revisor 1 |
|
|
The finding that self-identity and subjective norms negatively influence purchase intention contradicts most prior literature. The authors should provide stronger theoretical grounding (e.g., identity conflict, modernity vs. tradition) to explain this anomaly. A simple mention of “youth seeking modernity” is insufficient |
We thank the reviewer for their observation regarding the initial results. In response to this methodological concern, a second data collection was conducted, doubling the original sample size from 450 to 900 participants. Thus, the refined findings confirm the positive and significant influence of self-identity on purchase intention. Therefore, the initial inconsistency can be attributed to sample size limitations that affected the statistical stability of the model; however, expanding the sample yields more robust and reliable estimates, eliminating the apparent contradiction. You can see the new results starting from the line 223.
|
|
The use of convenience sampling undermines the representativeness of the findings. Please clarify why this method was chosen, and include a discussion on selection bias and limitations to external validity. |
We thank the reviewer for their comment. The reason for choosing this type of sampling has been explained in lines 192-202. Furthermore, in the section on limitations, the research bias of the sampling referred to in lines 588-594 has been acknowledged. |
|
The cross-sectional design limits the ability to infer causality. The authors should explicitly acknowledge this limitation and suggest longitudinal or experimental designs for future research. |
We thank the reviewer for their observation. This limitation has been stated in lines 595-603. |
|
The introduction does not adequately justify the focus on university students or their unique behavioral characteristics. |
We thank the reviewer for pointing out the need for a more robust justification for choosing university students as the target population. This observation has been addressed in lines 71-83. |
|
The literature review lacks a systematic synthesis of existing studies on healthy food purchase intentions, particularly among youth. |
We thank the reviewer for pointing out this important need to deepen the literature review. In response, we have synthesized previous studies on healthy food purchase intentions among young people, which can be seen in Table 1, line 193. |
|
All data were collected via self-reported questionnaires. A Harman’s single-factor test or latent method factor approach should be conducted to assess common method variance. |
Your comment has been taken into account. You can see it in lines 247 to 282. |
|
The theoretical model figure lacks clear labeling of hypothesized paths, reducing its explanatory value. |
We thank the reviewer for this observation. Figure 1 shows the labeling alongside their trajectories. You can see it on line 175.
|
|
Data collection via social media and teacher distribution may lead to common method bias and sample homogeneity. |
Thank you for this observation. We researchers have found it convenient to omit the common method step, given that the sample size is robust (n=900), a quantity that provides substantial statistical power to detect genuine effects and has the capacity to reduce sampling error. According to Monte Carlo simulations for SEM models (Wolf et al., 2013), samples larger than 500 cases allow for stable and reliable estimates even in the presence of minor assumption violations. |
|
Potential confounding variables (e.g., income, field of study) were not controlled for, affecting causal inference. |
We appreciate this observation. To remedy this, gender analysis has been taken into account, as shown in line 398-440.
|
|
The educational implications proposed in the conclusion are not directly derived from the results and lack empirical support. |
We appreciate this observation, which has allowed us to reformulate the implications, which can be seen in lines 555–585. |
|
The cross-cultural applicability of the findings, especially in other Latin American or developing contexts, is not discussed. |
We appreciate this observation, which has allowed us to take this important aspect into account. It has been incorporated into lines 453-460 and 471-474. |
|
Moral obligation is the strongest predictor, but its operationalization lacks theoretical depth and clarity. |
We appreciate this observation, which has allowed us to take this important aspect into account. It has been incorporated into lines 115-118 and 123-126.
|
|
The discussion does not sufficiently compare findings with similar studies from Asia or Africa, limiting its global relevance. |
We appreciate this observation, which has allowed us to take this important aspect into account. It has been incorporated into lines 453-460 and 471-474. |
|
The study did not analyze differences by gender or region, missing insights into segment-specific behaviors. |
We thank the reviewer for pointing out this important requirement. The analysis by gender has been carried out, as can be seen in line 398. |
|
The discussion section does not sufficiently interpret the negative effects of self-identity and subjective norms. A deeper psychological or sociocultural analysis is needed. |
Agradecemos la observación, luego de ampliar la muestra, se ha identificado variación en los resultados, aceptando todas las hipótesis de estudio |
|
Add a limitations subsection in the Discussion to clearly outline the study’s shortcomings. |
We appreciate the comment. For greater clarity, subsection 5.2, which identifies limitations and future research, has been modified. You can see this in lines 587-602 y 608-615 |
|
The policy and intervention implications are underdeveloped. Please elaborate on how these findings could inform university health programs, food marketing strategies, or public policy in Peru. |
We appreciate the comment. The section on implications has been reworded. You can see this in lines 55-585. |
|
The conclusions are too vague and lack specific, actionable recommendations for policy or marketing. |
We appreciate the comment. The conclusions section has been reworded. You can see this in lines 520-552 |
|
Reference formatting is inconsistent; some entries lack DOIs or page numbers(e.g., [15]). |
We appreciate the observation. The lack of DOIs in some cases is because they are not present in the document. |
|
Please proofread the manuscript for language and grammar issues, especially in the Introduction and Discussion sections. |
Thank you for your comment. The manuscript has been revised in English. |
|
Consider renaming “willingness to pay more” to “premium price willingness” for clarity and academic consistency. |
We appreciate the suggestion. After careful consideration, we have decided to retain “willingness to pay more” for the following conceptual reasons: • Semantic precision: The term “more” explicitly emphasizes the comparative nature of the construct (willingness to pay a premium/difference over conventional alternatives). • Operational clarity: In the measurement instrument, the items specifically asked about willingness to pay “more” compared to non-Andean grains (e.g., “I would be willing to pay more for Andean grains”), so the name of the construct accurately reflects the operationalization used. · • Consistency with previous literature: This terminology is consistent with previous studies analyzing the construct in question [1], [2], [3] |
|
There are two Table 1 in the manuscript. And Figures and tables should be reformatted for clarity and consistency with journal standards. |
Thank you for pointing that out. All tables have been revised. |
Reference
[1] Ş. Akan, E. Özdemir, and M. Bakır, “Purchase Intention Toward Green Airlines and Willingness to Pay More: Extending the Theory of Planned Behavior,” 2022, pp. 123–143. doi: 10.1007/978-981-16-9276-5_7.
[2] H. Sun et al., “The impact of brand authenticity on brand attachment, brand loyalty, willingness to pay more, and forgiveness - For Chinese consumers of Korean cosmetic brands -,” Heliyon, vol. 10, no. 16, p. e36030, Aug. 2024, doi: 10.1016/j.heliyon.2024.e36030.
[3] S. Siew, M. Minor, and R. Felix, “The influence of perceived strength of brand origin on willingness to pay more for luxury goods,” Journal of Brand Management, vol. 25, no. 6, pp. 591–605, Nov. 2018, doi: 10.1057/s41262-018-0114-4.
Reviewer 2 Report
Comments and Suggestions for Authors
The manuscript addresses a relevant topic, exploring the factors influencing purchase intention of traditional Andean grains among university students. The study is timely and potentially valuable, but in its current form it requires substantial revisions before it can be considered for publication.
The introduction is clearly structured and the hypotheses are well stated, for example H3 (attitude predicts purchase intention). However, the literature review is rather limited and does not sufficiently draw on recent studies published in the last 3–5 years that examine similar relationships in the context of functional or traditional foods. Strengthening the theoretical background would improve the justification of the hypotheses.
The methodology is described in a straightforward way, yet the characterization of the sample remains superficial. While Table 1 provides information on sex (301 women, 149 men), geographic origin (229 from the coast, 203 from the highlands, 18 from the jungle), and household status, important details such as mean age, academic program, or year of study are missing. This omission reduces the ability to assess representativeness. Moreover, since the sampling procedure was based on convenience, its implications for generalization should be explicitly acknowledged.
Regarding the statistical analysis, the use of PLS-SEM is appropriate, but the reporting is incomplete. For instance, high coefficients are reported (e.g., moral obligation β=0.728), but no confidence intervals are provided, nor is there a discussion of assumptions. Furthermore, there are issues of consistency: in several places the construct “attitude” (ATT) is mistakenly written as “ATK,” and the numbering of tables is duplicated (two different tables are labeled as Table 1). These errors should be carefully corrected.
The results themselves are interesting, particularly the unexpected finding that identity negatively influenced purchase intention, contrary to the initial hypothesis. However, the discussion does not sufficiently address this contradiction. Instead of critically engaging with the possible cultural or contextual reasons for this result, the text mainly repeats the findings. A deeper reflection on why student identity might play a negative role in this setting would considerably enrich the discussion.
The conclusion is somewhat generic, restating the hypotheses without presenting clear practical implications. The study would be stronger if the authors suggested strategies to promote the consumption of Andean grains among young people, for example through awareness campaigns or their integration into university food programs.
In addition, there are smaller issues to be addressed. The ethical approval code (2024-CEEPG-00158) is reported, which is commendable, but the process of informed consent should also be briefly described. Some tables contain redundant information that could be summarized in the text. The manuscript should also undergo a careful language revision to ensure consistency in terminology (e.g., “purchase intention” vs. “intention to purchase”) and to improve the academic style. Including comparative studies from other Andean countries or international markets could also help situate the findings more broadly.
In summary, the manuscript is promising, but requires significant revisions in terms of theoretical grounding, methodological clarity, statistical reporting, and depth of discussion. With these improvements, it could make a meaningful contribution to the literature on consumer behavior and traditional food systems.
Author Response
Ref.: Manuscript ID: foods-3895916
Title: Factors that influence the purchase intention of Andean grains in university students in the Peruvian market
Dear Reviewer 2,
Thank you very much for your valuable comments, which have been extremely helpful in improving our manuscript. We sincerely appreciate the time and effort you have dedicated to reviewing our work.
We have carefully addressed all your suggestions and made the necessary revisions accordingly. We hope that this revised version meets the expected standards and is now suitable for publication in Foods.
A detailed response to each of your comments is provided below.
Thank you again for your time and consideration.
Best regards,
Authors
|
Revisor 2 |
|
|
The introduction is clearly structured and the hypotheses are well stated, for example H3 (attitude predicts purchase intention). However, the literature review is rather limited and does not sufficiently draw on recent studies published in the last 3–5 years that examine similar relationships in the context of functional or traditional foods. Strengthening the theoretical background would improve the justification of the hypotheses. |
Thank you for your observation. In each hypothesis development, some important and updated guidelines have been added. You can see this in lines 92-96, 123-125, and 137-141. |
|
The methodology is described in a straightforward way, yet the characterization of the sample remains superficial. While Table 1 provides information on sex (301 women, 149 men), geographic origin (229 from the coast, 203 from the highlands, 18 from the jungle), and household status, important details such as mean age, academic program, or year of study are missing. This omission reduces the ability to assess representativeness. Moreover, since the sampling procedure was based on convenience, its implications for generalization should be explicitly acknowledged. |
Thank you for pointing that out. Age information has been included in lines 205 and 206. |
|
Regarding the statistical analysis, the use of PLS-SEM is appropriate, but the reporting is incomplete. For instance, high coefficients are reported (e.g., moral obligation β=0.728), but no confidence intervals are provided, nor is there a discussion of assumptions. Furthermore, there are issues of consistency: in several places the construct “attitude” (ATT) is mistakenly written as “ATK,” and the numbering of tables is duplicated (two different tables are labeled as Table 1). These errors should be carefully corrected. |
Confidence intervals have been considered in Table 5. See lines 359-398. |
|
The results themselves are interesting, particularly the unexpected finding that identity negatively influenced purchase intention, contrary to the initial hypothesis. However, the discussion does not sufficiently address this contradiction. Instead of critically engaging with the possible cultural or contextual reasons for this result, the text mainly repeats the findings. A deeper reflection on why student identity might play a negative role in this setting would considerably enrich the discussion. |
We thank the reviewer for their comment on the initial results. In response to this comment, a second data collection was conducted, doubling the original sample size from 450 to 900 participants. Thus, the refined findings confirm the positive and significant influence of self-identity on purchase intention. Therefore, the initial inconsistency can be attributed to sample size limitations that affected the statistical stability of the model; however, expanding the sample yields more robust and reliable estimates, eliminating the apparent contradiction. |
|
The conclusion is somewhat generic, restating the hypotheses without presenting clear practical implications. The study would be stronger if the authors suggested strategies to promote the consumption of Andean grains among young people, for example through awareness campaigns or their integration into university food programs. |
Thank you for your comment. The section on implications has been reworded. You can see it in lines 522-553. |
|
In addition, there are smaller issues to be addressed. The ethical approval code (2024-CEEPG-00158) is reported, which is commendable, but the process of informed consent should also be briefly described. Some tables contain redundant information that could be summarized in the text. The manuscript should also undergo a careful language revision to ensure consistency in terminology (e.g., “purchase intention” vs. “intention to purchase”) and to improve the academic style. Including comparative studies from other Andean countries or international markets could also help situate the findings more broadly. |
Thank you for your comment. This information is referred to in lines 632-635. |
|
In summary, the manuscript is promising, but requires significant revisions in terms of theoretical grounding, methodological clarity, statistical reporting, and depth of discussion. With these improvements, it could make a meaningful contribution to the literature on consumer behavior and traditional food systems. |
In the discussions, a comparison has been made 454-461 in other cross-cultural contexts 472-475. |
Round 2
Reviewer 1 Report
Comments and Suggestions for Authors
The formats of Tables (for example, the pagination issues of Tables 1, 2 and 3) and Figures (fonts) still need to be revised.
Reviewer 2 Report
Comments and Suggestions for Authors
The authors have addressed the majority of the previous comments with rigour and clarity. The revised manuscript presents a more robust theoretical background, improved methodological transparency, strengthened statistical reporting, and a more comprehensive and contextualised discussion of the findings. The structure is coherent, the tables are consistent, and the overall argumentation has been substantially enhanced.